# Investigations of High-Strength Mg–Al–Ca–Mn Alloys with a Broad Range of Ca+Al Contents

**DOI:** 10.3390/ma14185439

**Published:** 2021-09-20

**Authors:** Stefan Gneiger, Nikolaus P. Papenberg, Aurel R. Arnoldt, Carina M. Schlögl, Martin Fehlbier

**Affiliations:** 1LKR Light Metals Technologies Ranshofen, Austrian Institute of Technology, A-5282 Ranshofen, Austria; nikolaus.papenberg@ait.ac.at (N.P.P.); aurel.arnoldt@ait.ac.at (A.R.A.); carina.schloegl@ait.ac.at (C.M.S.); 2Chair of Foundry Technology, University of Kassel, Kurt-Wolters-Strasse 3, D-34109 Kassel, Germany; fehlbier@uni-kassel.de

**Keywords:** magnesium, alloy, extrusion, mechanical properties, microstructure

## Abstract

The low mass and high specific stiffness of Mg alloys make them particularly interesting as means of transportation. Due to further desirable properties, such as good machinability and excellent castability, Mg alloys have gained acceptance as castings in high-volume applications, such as gearbox housings and automotive steering wheels. However, in forming processes, such as extrusion and forging, Mg alloys find little to no industrial use at the moment. The reasons for this are their poor formability, which is reflected in limited processing speeds and low ductility, and their modest mechanical performance, compared to competing materials, such as Al alloys and high-strength steels. Much research is being conducted worldwide on high-strength Mg alloys, most of which rely on high levels of rare earths, making these materials both ecologically and economically questionable. Here, it is shown that high yield strengths (>300 MPa) can be achieved in the Mg–Al–Ca system while maintaining good ductility, using only low-cost elements. The investigations have shown that these properties can be adjusted over broad alloy compositions, which greatly simplifies both the processing and recyclability.

## 1. Introduction

Magnesium alloys can be very beneficial for automotive as well as aviation applications, where low weight of parts is crucial. While the casting of magnesium, especially high-pressure die casting, is widely used in the industry, wrought applications are still scarce. The reasons for this are the typical higher costs for extruded profiles and lower mechanical properties, when compared to aluminum. Nevertheless, while the mechanical properties achieved in casting processes are limited, high strengths can be achieved via extrusion.

High-strength magnesium extrusion alloys typically rely on at least one of the two main strengthening mechanisms: grain refinement, and/or precipitation hardening. Grain refinement is especially effective in Mg alloys, as the Hall–Petch constant is generally higher when compared to Al alloys [1]. As the strength increases with the inverse square root of the grain size, the effect is particularly pronounced at very low grain sizes (< 3 µm). Such small grain sizes in Mg alloys cannot be realized in conventional casting processes but are most effectively produced by dynamic recrystallization (DRX) occurring during forming processes. This can be most easily being achieved via severe plastic deformation (SPD) techniques. In the case of industrially established forming processes, i.e., extrusion or rolling, these small grain sizes are not as easily produced. It is nevertheless possible by using the right combination of alloy design and processing parameters. A relatively low deformation temperature generally leads to fine grains in the resulting material in deformation processes [2,3]. Additionally, the use of grain growth restricting intermetallic phases (IMP), which stabilize the microstructure by their applied Zener drag, allow for further improvement with regards to a fine and stable grain structure [4].

When investigating the works done on high strength Mg alloys, two major principal alloying groups can be identified: rare earth element (REE) containing alloys, and REE free materials. In the case of the alloys using REE, the achieved strengths are often dependent on the rare earth content of the alloy. The highest strengths are, therefore, achievable with high amounts of REE [5]. Because of their limited availability and high costs, the use of large amounts of REE as alloying elements can be considered unattractive for future industrial applications. 

While there have been multiple REE free alloying systems with high strength investigated [6,7], the system of Mg–Al–Ca is regarded as especially interesting to produce high strength components in industrial settings.

This system uses cheap and abundant alloying elements and offers a multitude of processing possibilities. Aluminum is the most important alloying element for Mg, both for casting- and wrought alloys, as it increases the castability and mechanical strength and enables the ability of precipitation hardening [8,9]. Calcium is used for grain refinement of the cast material [10,11]; it can weaken the texture in wrought material [12,13] and increase both the oxidation [14,15] and corrosion [16,17,18] resistance. Additionally, it allows the formation of IMPs with various structures (C36, C14, C15), depending on the Al- and Ca contents as well as the Ca/Al ratio [19,20,21]. Mn is often added to these alloys, as it binds Fe, thereby reducing the corrosion tendency [22]. Moreover, Al–Mn dispersoids inhibit grain growth during processing and solution heat treatment by Zener drag [23,24].

Matching the already described hardening mechanisms (precipitation hardening and grain refinement), two contrary processing trends for Mg–Al–Ca alloys can be found within the literature. One procedure uses the sequence of homogenization, forming and subsequent heat treatment steps, as known from age hardenable Al alloys, aiming to achieve high strengths in the final heat treatment (T6-temper) [25]. Since material strength is achieved by the formation of hardening phases, the number of precipitates formed during artificial aging is a decisive factor. Therefore, also alloys with a low overall alloying content, trimmed to the formation of these precipitates, can achieve high gains in final strength [23].

The other processing scheme is mainly based on the formation of small grains and the related increase in mechanical properties, such as hardness and ductility. In this case, heat treatments are not necessarily required, which reduces the processing steps and the associated disadvantages (energy consumption, material oxidation, grain growth tendencies, overaging, etc.) accordingly. Nevertheless, as the grain size must be minimized for the improvement of the mechanical properties, a high degree of deformation has to be achieved in a material with low grain growth tendencies. This is usually done at low extrusion temperatures and reduced forming speeds, demonstrating this principle for various alloying systems [7,26]. In the case of Mg–Al–Ca alloys, the works of Li et al. [20], Xu et al. [27] and Pan et al. [28] have provided exemplary material with outstanding mechanical properties in the as-extruded state, reaching a YS of 438, 410 and ~425 MPa, respectively. The alloys showed varying ductility of 2.5, 5.6 and 11% elongation to failure, nevertheless providing an impressive combination of strength and ductility. Li et al. investigated Mg–Al–Ca–Mn alloys with varying Ca/Al ratios, and reported the highest yield strength (438 MPa) for the alloy possessing the highest Ca/Al ratio (1.30) after extrusion at 350 °C [20]. As multiple alloying compositions and processing temperatures have been investigated in the scientific literature, a graphical overview of the data obtained from sources [20,27,28,29,30,31] is presented in Figure 1. For easier understanding, the experimental data are additionally listed in Table A3 (Appendix A).

As can be seen in Figure 1, the ratio of Ca/Al and the processing temperatures seem to have a direct influence on the mechanical properties of the as-extruded Mg–Al–Ca alloys. It is noteworthy that high YS values were consistently achieved at a Ca/Al ratio (in wt.%) of ~1 over a wide range of processing temperatures and total alloy content.

For this behavior, we investigated a wide range of Mg–Al–Ca–Mn alloys processed with identical processing parameters to improve comparability, focusing on alloys with an Ca/Al ratio of ~1 with the sum of Al+Ca content ranging from 1 to 10 wt.%. All alloys shown were processed by indirect extrusion at 350 °C. Both microstructural and mechanical properties were analyzed, and the results shown should help to further promote understanding of this interesting alloying system.

## 2. Materials and Methods

A variation of Mg–Al–Ca–Mn alloys with a Ca/Al ratio of ~1 and a total Al+Ca content of ~1 wt.% to ~10 wt.% was used for the studies. The nominal chemical compositions of the alloys are given in Table 1. The alloys were produced in a resistance heated furnace (Nabertherm Liquitherm^®^ T20/H, Lilienthal, Germany) in a mild steel crucible with a melt capacity of 30 kg, using commercially pure Mg (99.8%) and Al (99.85%), Ca granules (99.9%) and MnCl_2_ flakes (98%). Ar with 1 vol.% SF_6_ was used as a cover gas during the melting and handling.

The alloys were gravity cast into cylindrical billets with a diameter of 65 mm and a height of 240 mm in a tempered (approx. 150 °C) mild steel mold. The billets were then turned to a diameter of 48.5 mm and a length of 180 mm for subsequent extrusion. No homogenization heat treatment was performed prior to extrusion.

The billets were then inductively heated and indirect hot extruded, using a 1.5 MN direct extrusion press (NEHP 1500.01, Müller Engineering, Todtenweis, Germany) to circular profiles with a diameter of 10 mm at 350 °C with an extrusion ratio of 25:1. The ram speed was 0.45 mm/s, which resulted in a die exit speed of 0.70 m/min. The profiles were left on the conveyor of the extrusion press until fully cooled. No additional heat treatment was performed.

After extrusion, tensile test samples acc. to DIN 50125 A8 × 40 were manufactured out of the profiles, and at least five samples each were tested at room temperature, using a Zwick Z100 tensile testing machine (Zwick Roell, Ulm, Germany). Microstructural investigations were carried out using optical light microscopy and scanning electron microscopy (SEM). A Tescan MIRA3 field emission gun (FEG) scanning electron microscope (Brno, Czech Republic) operated at 20 kV and a working distance of 15 mm was used for imaging, energy dispersive X-ray spectroscopy (EDS) and electron backscattered diffraction (EBSD). The images were taken using a 4-quadrant solid-state backscattered electron (BSE) detector. The element analysis was performed with an EDAX (Mahwah, NJ, USA.) Octane Elect silicon drift detector system, and an electron backscattered diffraction (EBSD) was performed using an EDAX Hikari Plus EBSD camera. EBSD data analysis was carried out with EDAX OIM Analysis 8 (Mahwah, NJ, USA). For cleanup, we used one iteration of grain dilatation with a grain tolerance angle of 5° and a minimum grain size of 2 px and neighbor CI correlation with a min. CI of 0.1. For the grain size analysis, a misorientation angle of 15° was used to determine the grain boundaries. 

For a measurement of the recrystallized grain fraction, the recrystallized and non-recrystallized grains were divided by their grain orientation spread (GOS) as discussed by [32].

Thermodynamic calculations were made using Thermo-Calc software (Version 2021b) and databases TCMG5 and TCMG6 (Thermo-Calc Software AB, Solna, Sweden). The calculations were performed, using average values of the compositions given in Table 1.

## 3. Results

To provide an in-depth analysis of the produced alloys, microstructural investigations using light microscopy as well as SEM/EBSD were performed, and CALPHAD calculations were used to obtain additional information on the phase formation. The mechanical properties were evaluated by tensile tests at room temperature. 

### 3.1. Thermodynamic Calculations

To allow for a better understanding of the phase formation in the alloys used in this work, the respective thermodynamic calculations in equilibrium state are shown in Figure 2. Additionally, an isothermal section at 350 °C (extrusion temperature) of the ternary phase diagram for 0–10 wt.% of Ca and Al is given.

In binary Mg–Ca alloys, the prevalent phase is the C14-structured Mg_2_Ca, which forms either directly from the liquid or precipitates from the supersaturated solid. In ternary Mg–Ca–Al alloys, Al_2_Ca, (Mg,Al)_2_Ca and Mg_2_Ca can form, depending on the local Ca/Al ratio [33,34,35].

All investigated alloys possess a Ca/Al ratio of ~1 (in wt.%) but differ in the total amount of Ca+Al. Since the Mn present in the alloys forms Al_8_Mn_5_ during solidification before reactions between Ca and Al occur, the effective Ca/Al ratio increases as Al is consumed. This effect is more pronounced in alloys with a lower Ca+Al content, when compared with the higher alloyed ones.

In the alloys with a higher Mn content (AX22, AX43 and AX55), Al_8_Mn_5_ is formed before α-Mg, while in AX01 and AX11, the temperature of formation is lower than the formation of α-Mg. As the Al_8_Mn_5_ phases form directly from the melt, the solidification speed is critical to the size of the Al–Mn particles, especially as they cannot be dissolved during the subsequent processing steps.

The amount of formed Ca containing phases rises correspondingly to the alloying content, ranging from a phase fraction of 0.9% up to 8.6% (at 350 °C). In all alloys, except for AX01 and AX11, the C36 (Mg,Al)_2_Ca phase forms directly out of the melt at eutectic temperature. With reduced alloying content, the phase formation is shifted toward lower temperatures. Therefore, in AX11, it precipitates from the supersaturated solid, while in AX01, the C36 phase does not form. On the other hand, the temperatures of phase formation for C14 (Mg_2_Ca) and C15 (Al_2_Ca) are independent of the alloying content, except for AX01 (Figure 2a).

At extrusion temperature (350 °C), the calculations indicate that no (Mg,Al)_2_Ca is present in the equilibrium state and only Mg_2_Ca, Al_2_Ca and Al_8_Mn_5_ can be found in addition to the Mg solid solution (Figure 2f). Nevertheless, as no heat treatment was performed on the investigated alloys prior to extrusion, it cannot be excluded that remnants of this phase are still present.

It should be noted that, under real casting conditions, in contrast to the equilibrium calculations shown, phase formation during solidification may be shifted toward lower temperatures, due to the local solidification rate. This is particularly noteworthy because the materials in this study were extruded without prior heat treatment, and thus a non-equilibrium state has to be assumed. In this case, phases can additionally settle out of the supersaturated matrix during heating/extrusion.

### 3.2. Microstructure

In the as-cast microstructure (Figure 3), lamellar eutectic, Ca containing phases can be found on the grain boundaries in all investigated alloys, with its total amount dependent on the respective alloying content. Furthermore, block shaped Al–Mn phases are present in all alloys. As described in Section 3.1, the alloys with the higher Mn content form larger primary Mn particles, reaching sizes of up to 10 µm, while in the alloys with lower Mn content, the phases are accordingly smaller. 

Figure 4 shows the microstructure of alloy AX11 in the as-extruded state in extrusion (ED) and transverse directions (TD). Ca-rich compounds as well as Al–Mn compounds can be seen. The Al–Ca compounds stemming from as-cast microstructure are fragmented and arranged in rows along the extrusion direction. A small number of the Al–Ca compounds remain unaffected during deformation and retain their spherical shape.

The grain size of the as-cast materials is given in Figure 5a and an overview of the average sizes of the DRXed grains in the as-extruded material are given in Figure 5b. In contrast to the widely differing as-cast grain size, the average grain sizes of all extruded alloys are in the same range, roughly between 1 and 2 µm. With increasing Ca+Al content, the grain size in the as-extruded material (up to approx. 6.5 wt.% Ca+Al) decreases steadily, while it increases again in the AX55 alloy. 

Figure 6 shows the grain orientation spread (GOS) for all extruded alloys, separately analyzed for the profile center and rim. The as-extruded alloys exhibit a bi-modal microstructure, consisting of small DRXed grains and large deformed structures. The DRXed grains are defined by a GOS value < 2 and are the predominant area fraction within the measured samples. In general, the sample areas consist of 80 to 90% recrystallized grains at the profile rim, while the center has a recrystallized area of 55–80%. As the EBSD measurements were done in ED, the fraction of not recrystallized (nDRX) grains frequently displays deformed, curving grains, which are well comparable to the warped (particle) structures depicted in Figure 4a. Details on the respective area fractions of recrystallized grains can be found in Table A1 (Appendix A).

### 3.3. Texture Measurements of As-Extruded Alloys

The pole figures depicted in Figure 7 show a basal texture typical for as-extruded Ca containing Mg alloys, i.e., <10–10> planes oriented parallel to ED. When comparing sample centers with the rim measurements, a distinctive higher texture intensity can be seen. This effect is especially pronounced in the alloys AX22 and AX43, where the texture intensity of the center is more than double the value of that of the sample rim. 

### 3.4. Mechanical Properties

The mechanical properties of the as-extruded materials obtained from tensile tests performed at room temperature are given in Figure 8. All alloys show good mechanical properties with YS of >285 MPa and UTS values greater than 315 MPa while still boasting a minimum elongation at break of 5.5%. The exact values can also be found in Table A2 in Appendix A.

The yield strength as well as the ultimate tensile strength increase continuously with increasing the content of Ca+Al up to AX43, which shows the highest strength of all investigated alloys. The elongation at break decreases with increasing the alloying content, where AX43 possesses the lowest value. For higher alloying contents, these trends reverse, and the strength decreases while the elongation is slightly improved, compared to AX43. Therefore, AX55, although it contains the highest amounts of Ca+Al, has strength values comparable to AX22 but with a lower elongation at break.

### 3.5. Thermal Stability

To assess the thermal stability of the alloys, AX11 was investigated by annealing treatments, reflecting potential applications. AX11 was chosen based on its overall good combination of strength and elongation at break. Additionally, the relatively low combined content of Al+Ca might lead to increased temperature response and, therefore, accelerated grain growth. The extruded material was exposed to temperatures of 150 °C and 250 °C for 9 and 24 h, respectively. The results of the tensile tests can be seen in Table 2. In comparison to the as-extruded material, the YS and the UTS decrease slightly, while the elongation at break remains stable.

## 4. Discussion

As shown in Figure 1, alloys with a Ca/Al ratio of <0.9 possess lower mechanical properties than alloys with a Ca/Al ratio of >0.9 when processed at the same parameters. This coincides well with the various phase regions possible in Mg–Al–Ca alloys. Materials that have a Ca/Al-ratio of >1 are in the phase region α+C14, while materials with a Ca/Al-ratio < 1 can be found in the region α+C15. The highest mechanical properties reached at various processing temperatures were found in alloys with a Ca/Al-ratio of ~1, which are in the phase region α+C14+C15. As can be seen in Figure 2, all investigated alloys fit into this specific criterion. 

The microstructure found in the as-extruded material is typical for this kind of alloy and is described in detail by [27]. The fine grain sizes obtained during extrusion stem from of dynamically recrystallized grains, which are limited in their growth by fractured Ca containing phases and Al–Mn dispersoids. In combination with the non-recrystallized grains, a bi-modal microstructure is obtained. This fraction of the non-recrystallized grains (nDRX) increases from profile rim towards the center of the extruded profiles. This is most likely caused by the increase in temperature and degree of deformation at the sample rim during the extrusion process.

The measured texture is common for as-extruded Mg–Al–Ca alloys, i.e., a basal fiber texture, as reported in [27]. The texture is comprised of both, DRX and nDRX grains, but it can be assumed that the intensity is strongly influenced by the nDRXed structures. This is visible in the difference of texture intensity between the profile center and rim. The rim shows a considerable reduction in texture intensity, as it consists of a larger fraction of DRXed grains. While the texture is distinct, the maximum intensities are still acceptable when compared to other as-extruded Mg–Alloys, e.g., AZ31 [36,37]. This reflects the large impact of Ca on the texture development in these alloys. The number and size of the DRXed grains coincides fairly well with the number of intermetallic phases in the alloys, which can indicate an effect of particle stimulated nucleation on the DRXed fraction. 

There are multiple reasons for the good mechanical properties of this alloy system. Regardless of the initial cast grain size, a fine microstructure is obtained after forming. These small grain sizes within the DRXed areas provide good ductility as well as high strengths by grain boundary hardening (Hall–Petch). Additionally, the nDRXed grains of this bi-modal structure are an important feature, which can directly influence the resulting YS by sub-grain hardening, as reported in [28]. Additionally, the basal fiber texture (which is strongly influenced by the number of nDRXed grains in the microstructure) can be responsible for a reduced Schmid factor in the case of a basal slip when deformed in the extrusion direction. As the described basal structure disproportionately improves the mechanical strength in extrusion direction, the material can be assumed to have anisotropic properties. This effect is most likely pronounced in this investigation, as the tensile tests shown were performed at room temperature and in the extrusion direction. Nevertheless, this is a common approach when investigating extrusion alloys at laboratory scale as can be seen in the cited literature.

As aging treatments show little effect on the mechanical properties of this type of as-extruded Mg–Al–Ca–Mn alloy [30,31], components that are intended for use at elevated temperatures, e.g., powertrain applications, could be considered. The Ca and Mn containing phases found in the alloys are thermally stable up to temperatures of ~350 °C (Figure 2) while still retaining the desired grain structure.

## 5. Conclusions

We investigated the microstructure and mechanical properties of extruded Mg–Al–Ca–Mn alloys with a Ca/Al ratio of ~1 (in wt.%) and a total Ca+Al content ranging from ~1 wt.% up to 10 wt.%, and achieved the following results:By maintaining a constant Ca/Al ratio, a wide selection of alloys with similar structure and attractive mechanical properties (YS > 285 MPa, elongation > 5.5%) can be created.The combination of extrusion process and intermetallic phase formation results in a bi-modal grain structure with a large fraction (up to 89%) of fine recrystallized grains (<2 µm). The intermetallic phases enable this fine grain structure by inhibiting grain growth and facilitating recrystallization.Alloy production can be realized with the use of abundant raw materials (Mg, Al, Ca, and Mn).No heat treatments are necessary during processing.The obtained mechanical properties are well maintained after exposure for 24 h at elevated temperatures (250 °C), as the intermetallic phases present are thermally stable.No narrow tolerances for chemical composition (except a Ca/Al ratio of >0.9) need to be maintained; therefore, alloy production and recycling are simplified.

Concluding, we believe that this advantageous combination of properties could arouse industrial interest and enable future applications of Mg–Al–Ca–Mn alloys.

## Figures and Tables

**Figure 1 materials-14-05439-f001:**
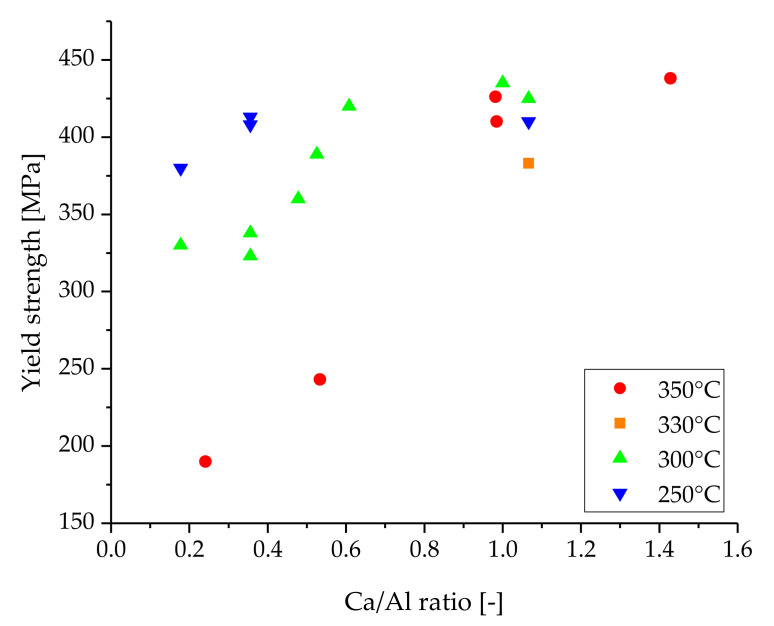
Overview of as-extruded Mg–Al–Ca alloys investigated in the scientific literature. The Ca/Al ratio (in wt.%) of these alloys is correlated to their tensile yield strength and the respective forming temperature.

**Figure 2 materials-14-05439-f002:**
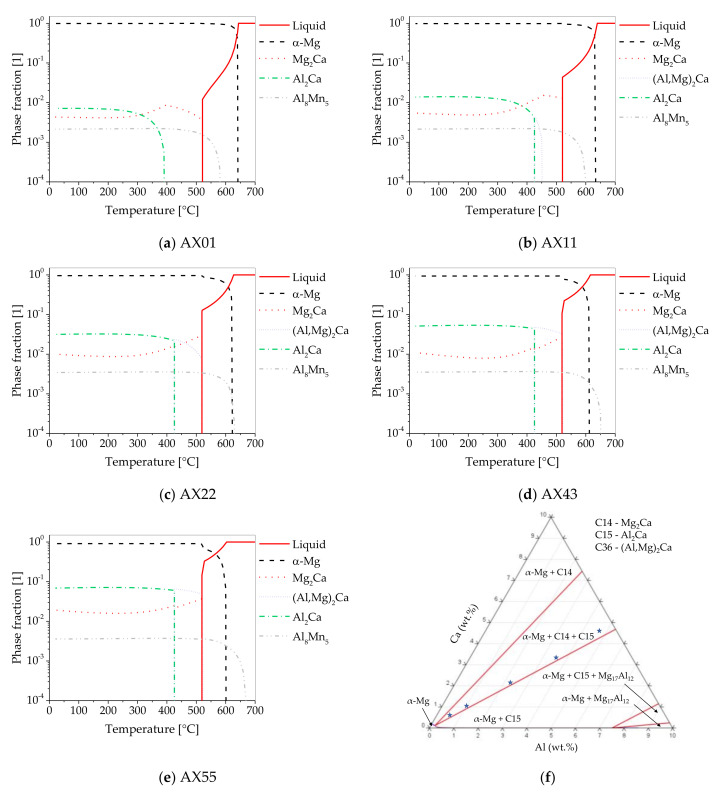
Thermodynamic equilibrium calculations (**a**–**e**). (**f**) Ternary Mg–Al–Ca phase diagram (isothermal section at 350 °C), including the stable phases, investigated alloy compositions are marked.

**Figure 3 materials-14-05439-f003:**
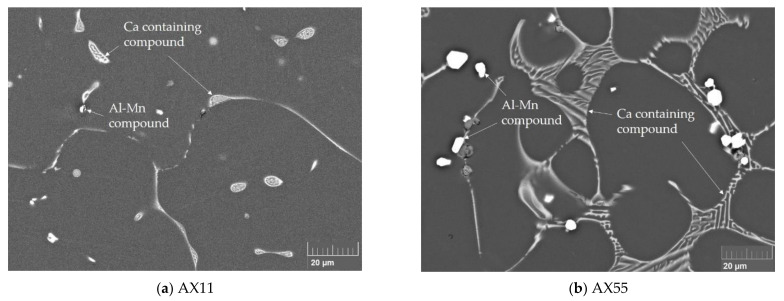
As-cast microstructures of AX11 (**a**) and AX55 (**b**).

**Figure 4 materials-14-05439-f004:**
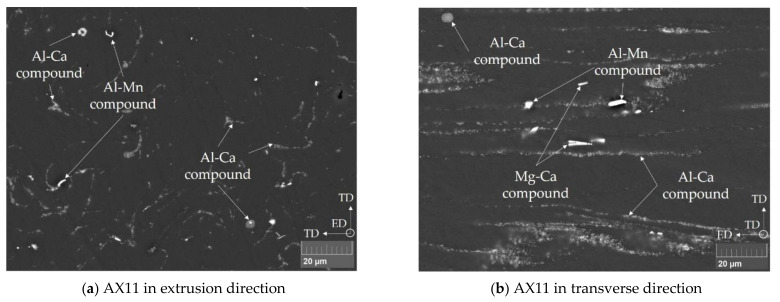
SEM micrographs of alloy AX11 in (**a**) extrusion direction (ED) and in (**b**) transverse direction (TD). Ca-rich compounds as well as Al–Mn compounds can be seen.

**Figure 5 materials-14-05439-f005:**
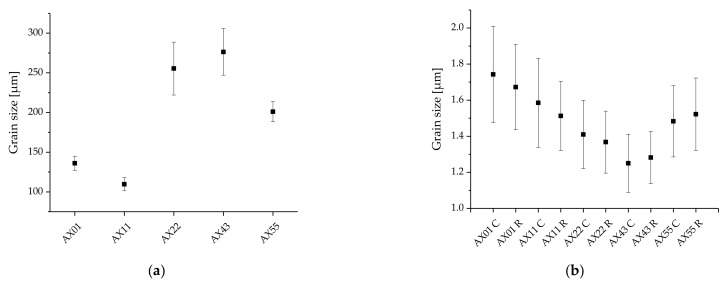
Grain size of the as-cast billets (**a**) and grain size of the recrystallized areas in the extruded material (**b**). In the case of the extruded material, the analysis was conducted separately for the profile center (C) and the profile rim (R). Error bars show 95% confidence interval.

**Figure 6 materials-14-05439-f006:**
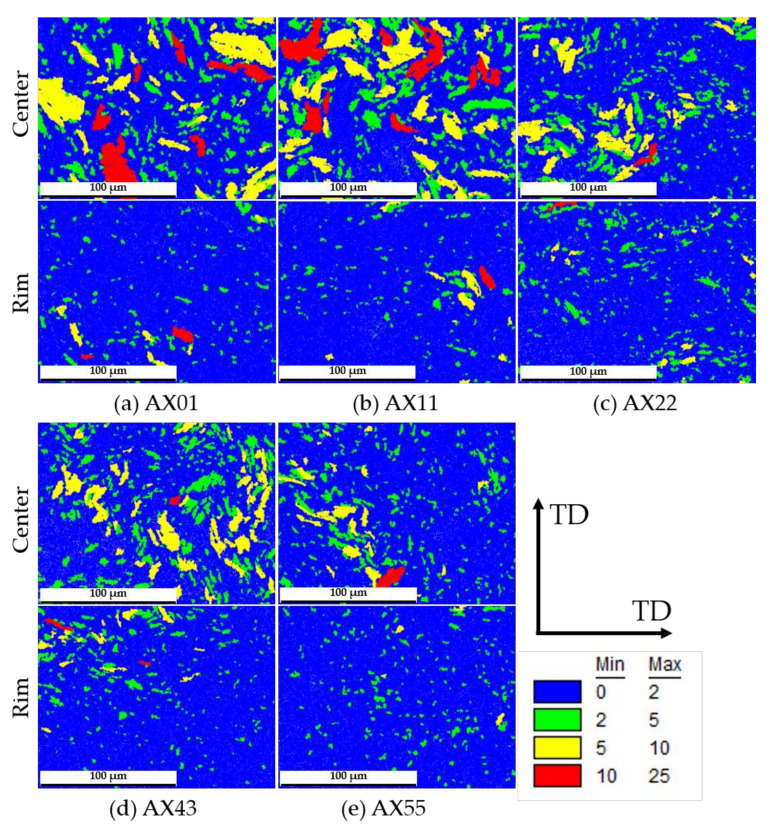
Grain orientation spread (GOS) in extrusion direction; grains sorted by their GOS values; the fraction with GOS values <2 can be assumed to be recrystallized.

**Figure 7 materials-14-05439-f007:**
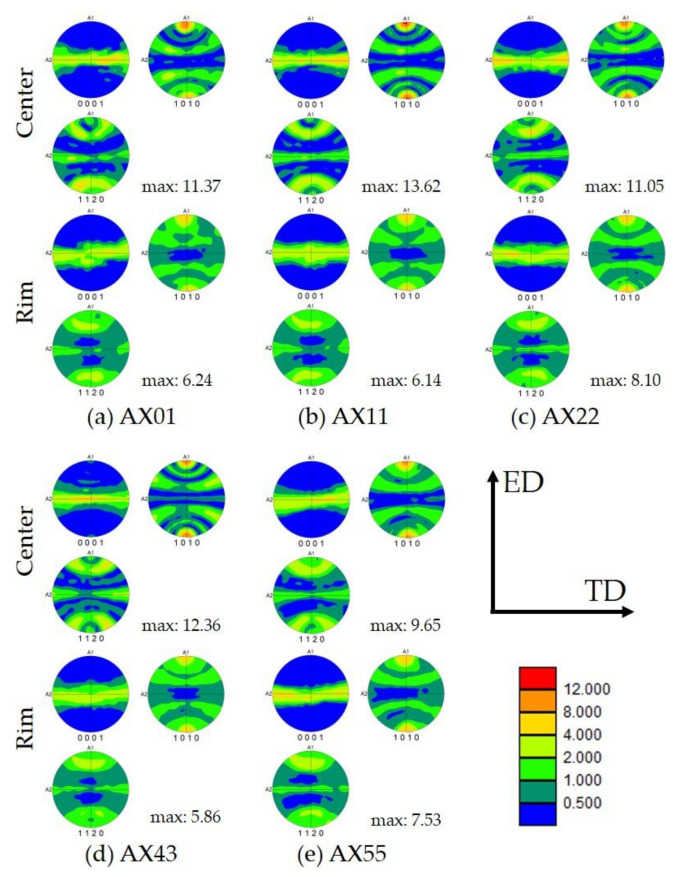
Pole figures of the as-extruded alloys. Figures given are (0001), (10–10) and (11–20).

**Figure 8 materials-14-05439-f008:**
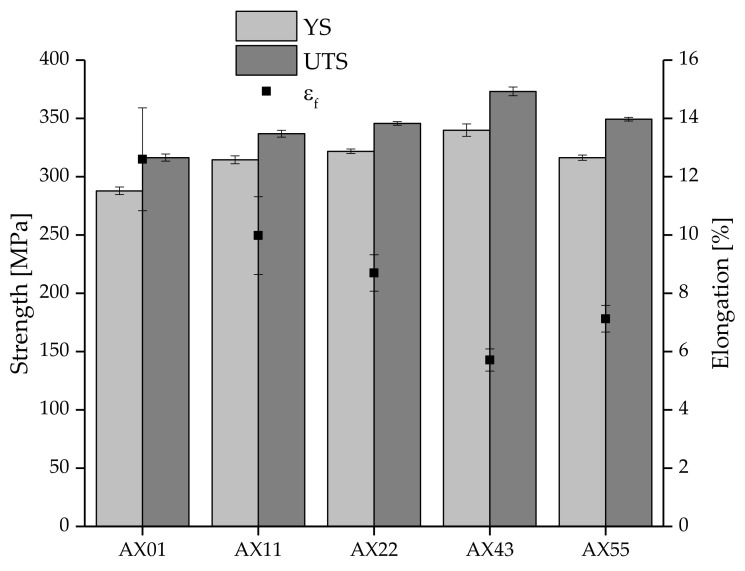
Tensile test results (room temperature) of all investigated alloys in the as-extruded state.

**Table 1 materials-14-05439-t001:** Nominal alloying compositions.

Alloy	Al [wt.%]	Ca [wt.%]	Mn [wt.%]	Mg	Ca/Al Ratio * [-]
AX01	0.50–0.70	0.55–0.70	0.20–0.30	Balance	1.04
AX11	1.00–1.10	1.00–1.10	0.20–0.30	Balance	1.00
AX22	2.20–2.40	2.10–2.40	0.35–0.45	Balance	0.98
AX43	3.40–3.80	3.20–3.50	0.35–0.46	Balance	0.93
AX55	4.50–5.00	4.40–5.00	0.35–0.47	Balance	0.99

* Ca/Al ratio based on the average alloy compositions in wt.%.

**Table 2 materials-14-05439-t002:** Mechanical properties (tensile at RT) of AX11, as extruded and after annealing treatment.

Condition	YS [MPa]	UTS [MPa]	ε_f_ [%]
As extruded	314.5 ± 3.4	336.9 ± 3.0	10.0 ± 1.3
150 °C/9 h	304.8 ± 4.5	326.2 ± 3.4	11.4 ± 0.8
250 °C/24 h	288.9 ± 1.3	321.4 ± 0.7	9.7 ± 1.6

## Data Availability

The raw/processed data required to reproduce these findings cannot be shared at this time as the data also form part of an ongoing study.

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
