# Peer review of "Investigations of High-Strength Mg–Al–Ca–Mn Alloys with a Broad Range of Ca+Al Contents"

_materials, 2021, doi:10.3390/ma14185439_

Round 1

Reviewer 1 Report

  • Author should clearly describe novelty of this manuscript in the end of Introduction section.
  • Font of text in Fig. 2 is too small and it is not good readable.
  • Scales in Fig. 6 is not readable at all.
  • Author should add some pictures and description of shape of a ductile specimens for tensile test. 
  • Tensile test have to be improved and described better. There should be shown more results of this test and author should add fracture surfaces with comment and description.
  • Author should enrich chapter 3.3 with more description and comments to results of texture measurements.
  • Author should add picture of samples.

Author Response

Thank you for taking your time to review our article in detail and giving your valuable comments. We have improved the article based on your suggestions.

  • Author should clearly describe novelty of this manuscript in the end of Introduction section.

We improved the last paragraph of the Introduction section.

  • Font of text in Fig. 2 is too small and it is not good readable.

Thank you for pointing this out – there was a problem with inserting the picture into the word-file; therefore, the text was not readable very well. We exchanged all pictures in the figure and now used .emf data format. The font size was increased, sharpness was enhanced, and line thickness was also increased.

  • Scales in Fig. 6 is not readable at all.

We changed the Figure / added bigger scale bars so that the scales are now readable.

  • Author should add some pictures and description of shape of a ductile specimens for tensile test. 
  • Tensile test have to be improved and described better. There should be shown more results of this test and author should add fracture surfaces with comment and description.
  • Author should add picture of samples.

Tensile tests were performed according to standard DIN EN ISO 6892-1 and tensile test geometry was according to standard DIN 50125 A8x40.

As the samples were manufactured according to a standard and the exact definition is given in the “Materials and Methods” section, we don’t see the necessity to add a picture of a tensile test sample as it can be found in the specified standard.

5 samples from each alloy were tested and no samples were omitted from the results. As the standard deviation is low, this number of samples is considered sufficient. More samples would not increase the informative value.

Nevertheless, in addition to the diagram already included in the article, a table with the values was added in the appendix (Table A2).

  • Author should enrich chapter 3.3 with more description and comments to results of texture measurements.

In our opinion, the important results of the texture measurements were described in the text in chapter 3.3. Further interpretation of this results can be found in chapter “Discussion”. If in your opinion we have missed something important here, please let us know in detail.  

Reviewer 2 Report

The research article “Investigations of high-strength Mg-Al-Ca-Mn alloys with a broad range of Ca+Al contents” is essential for deeply understanding the microstructure and mechanical properties of extruded Mg-Al-Ca-Mn alloys. Authors show that high yield strengths (> 300 MPa) can be achieved in the Mg-Al-Ca system while maintaining good ductility using only low-cost elements. The investigations have shown that these properties can be adjusted over broad alloy compositions, which greatly simplifies both processing and recyclability. This manuscript is dedicated to behavior of Ca/Al ratio on properties. A wide range of Mg-Al-Ca-Mn alloys produced by extrusion at 350°C, focusing on alloys with an Ca/Al ratio of ~ 1 with the sum of (Al+Ca) content ranging from 1 to 10 wt. % is observed. My recommendation is to accept the article in present form.

Author Response

Thank you for your valuable comment and the recommendation to accept the article in its present form. We are very pleased that you took the time to review the article and that you liked it.

Reviewer 3 Report

The research paper “Investigations of high-strength Mg-Al-Ca-Mn alloys with a broad range of Ca+Al contents” is an interesting study in the area of Mg-based alloys. I recommend this paper for publishing in Materials after the required correction.

Comments:

  1. Figure 3. The SEM images of as-cast structure and EDS-analysis are required to confirm the presence of the indicated phases.
  2. How was calculated the grain size after extrusion? The grains are not obvious in figure 4. The analysis of grain structure after extrusion is required.
  3. Why have the authors chosen the alloy AX11 for thermal stability analysis (Section 3.5), whereas the highest properties were obtained for alloy AX43?
  4. The draw conclusions (section 5) are too superficial. For example, “By maintaining a constant Ca/Al ratio, a wide selection of alloys with similar structure and attractive mechanical properties can be created.” The “attractive properties” is a subjective meaning that is not of scientific significance. I recommend authors provide exact values or limits for measured characteristics. The same recommendations can be addressed to other conclusions. “Large fraction of recrystallized grains”, “fine grain”, “high mechanical properties”, “elevated temperatures”, “narrow tolerance of composition” etc. These parameters should be summarized and quantitively described in conclusions sections as concrete values or intervals.

Author Response

The research paper “Investigations of high-strength Mg-Al-Ca-Mn alloys with a broad range of Ca+Al contents” is an interesting study in the area of Mg-based alloys. I recommend this paper for publishing in Materials after the required correction.

Thank you for taking your time to review our article in detail and giving your valuable comments. We agree with your comments and made the recommended changes to improve the article.  

Comments:

  1. Figure 3. The SEM images of as-cast structure and EDS-analysis are required to confirm the presence of the indicated phases.

Exchanged the optical light microscope images with SEM images. EDS analysis were performed to identify the phases but are not shown in the article.

  1. How was calculated the grain size after extrusion? The grains are not obvious in figure 4. The analysis of grain structure after extrusion is required.

The grain size after extrusion was analyzed via EBSD as has been described in the experimental section:  

“EBSD data analysis was carried out with EDAX OIM Analysis 8. For cleanup, we used one iteration of grain dilatation with a grain tolerance angle of 5 ° and a minimum grain size of 2 px and neighbor CI correlation with a min. CI of 0.1. For grain size analysis, a misorientation angle of 15 ° was used to determine grain boundaries.

For a measurement of the recrystallized grain fraction, the recrystallized and non-recrystallized grains have been divided by their grain orientation spread (GOS)…”

  1. Why have the authors chosen the alloy AX11 for thermal stability analysis (Section 3.5), whereas the highest properties were obtained for alloy AX43?

The alloy AX11 was chosen based on its overall good combination of strength and elongation at break. Additionally, the total alloying content of AX11 is significantly lower than in AX43, therefore the amount of thermally stable phases is also smaller. Therefore, it can be assumed when AX11 shows good thermal stability, the stability of AX43 should be at least equal or better.

We added an explanation in the paragraph of section 3.5.

  1. The draw conclusions (section 5) are too superficial. For example, “By maintaining a constant Ca/Al ratio, a wide selection of alloys with similar structure and attractive mechanical properties can be created.” The “attractive properties” is a subjective meaning that is not of scientific significance. I recommend authors provide exact values or limits for measured characteristics. The same recommendations can be addressed to other conclusions. “Large fraction of recrystallized grains”, “fine grain”, “high mechanical properties”, “elevated temperatures”, “narrow tolerance of composition” etc. These parameters should be summarized and quantitively described in conclusions sections as concrete values or intervals.

Conclusions were reworked, and exact values were added.

Round 2

Reviewer 1 Report

Author reflected all my comments and improved his manuscript.